# Patient-derived tumor organoids for personalized medicine in a patient with rare hepatocellular carcinoma with neuroendocrine differentiation: a case report

Marie-Anne Meier[1,2,8], Sandro Nuciforo [1,8], Mairene Coto-Llerena[1,3,8], John Gallon [1], Matthias S. Matter [3], Caner Ercan [3], Jürg Vosbeck[3], Luigi M. Terracciano[4,5], Savas D. Soysal[2], Daniel Boll[6], Otto Kollmar[2], Raphaël Delaloye[7], Salvatore Piscuoglio [1,3✉] & Markus H. Heim [1,2✉]

## Abstract

**Background** Hepatocellular carcinoma with neuroendocrine differentiation (HCC-NED) is a very rare subtype of primary liver cancer. Treatment allocation in these patients therefore remains a challenge.

**Methods** We report the case of a 74-year-old man with a HCC-NED. The tumor was surgically removed in curative intent. Histopathological work-up revealed poorly differentiated hepatocellular carcinoma (Edmondson-Steiner grade IV) with diffuse expression of neuroendocrine markers synaptophysin and chromogranin. Three months after resection, multifocal recurrence of the HCC-NED was observed. In the meantime, tumor organoids have been generated from the resected HCC-NED and extensively characterized. Sensitivity to a number of drugs approved for the treatment of HCC or neuroendocrine carcinomas was tested in vitro.

**Results** Based on the results of the in vitro drug screening, etoposide and carboplatin are used as first line palliative combination treatment. With genomic analysis revealing a *NTRK1*-mutation of unknown significance (kinase domain) and tumor organoids found to be sensitive to entrectinib, a pan-TRK inhibitor, the patient was treated with entrectinib as second line therapy. After only two weeks, treatment is discontinued due to deterioration of the patient's general condition.

**Conclusion** The rapid establishment of patient-derived tumor organoids allows in vitro drug testing and thereby personalized treatment choices, however clinical translation remains a challenge. To the best of our knowledge, this report provides a first proof-of-principle for using organoids for personalized medicine in this rare subtype of primary liver cancer.

## Plain language summary

Tumors that simultaneously display features of liver, nerve and hormone-producing cells are very rare. In such cases, the most appropriate treatment choice is not well defined. Here, we describe the generation of three-dimensional miniature tumors, called organoids, from the patient's tumor tissue, that can be grown and studied in a culture dish. These organoids closely mimic the patient's tumor and allowed us to test different drugs to identify the most effective therapy for informed treatment choice. What we describe in this study is an emerging approach for a practice known as personalized medicine, that aims to provide a more tailored treatment to patients. In summary, we demonstrate that this approach can be useful in a rare cancer type and that it holds significant potential to guide treatment decision in other patients with aggressive cancers.

[1] Department of Biomedicine, University Hospital and University of Basel, CH-4031 Basel, Switzerland. [2] Clarunis University Center for Gastrointestinal and Liver Diseases, CH-4002 Basel, Switzerland. [3] Institute of Medical Genetics and Pathology, University Hospital Basel, CH-4031 Basel, Switzerland. [4] Department of Anatomic Pathology, IRCCS Humanitas Research Hospital, Rozzano, Milan, Italy. [5] Humanitas University, Department of Biomedical Sciences, Pieve Emanuele, Milan, Italy. [6] Radiology and Nuclear Medicine, University Hospital Basel, CH-4031 Basel, Switzerland. [7] Department of Oncology, University Hospital Basel, CH-4031 Basel, Switzerland. [8] These authors contributed equally: Marie-Anne Meier, Sandro Nuciforo, Mairene Coto-Llerena. ✉email: s.piscuoglio@unibas.ch; markus.heim@unibas.ch

Primary liver carcinomas with concurrent hepatocellular and neuroendocrine tumor components in the same liver lesion are very rare[1]. They consist of two morphologically distinct cell populations that express hepatocellular or neuroendocrine markers and are classified as Hepatocellular Carcinoma-Neuroendocrine Carcinoma (HCC-NEC)[2] or liver mixed neuroendocrine non-neuroendocrine neoplasms (MiNEN)[3]. The published case reports describe an aggressive tumor phenotype and poor overall prognosis[3,4]. Even rarer are HCCs with neuroendocrine differentiation (HCC-NED)[5]. HCC-NEDs are comprised of morphologically uniform cells that stain positively for both hepatocellular and neuroendocrine markers. Patients are usually treated by means of surgical resection, transarterial chemoembolization or systemic (chemo)therapy for liver cancer or neuroendocrine malignancies. Because HCC with neuroendocrine differentiation is a very rare tumor entity, therapy in these patients remains ill-defined[4,6,7]. Here, we report a case history of a 74-year-old man with HCC-NED. We provide a comprehensive histopathological characterization and a genomic analysis of this rare tumor. Furthermore, we describe the generation of tumor organoids that retain the key characteristics of the originating tumor. The organoids were used in drug screens to identify the most promising treatment options.

## Methods

**Patient information and biological material**. Human biopsy and resection tissue was collected from patients undergoing diagnostic liver biopsy or liver surgery at the University Hospital of Basel. Written informed consent was obtained from all patients. The study was approved by the local ethics committee (protocol numbers EKNZ 2014-099 as well as BASEC 2019-02118). For the HCC-NED patient described in this study, written informed consent to publish the case details was obtained from the family.

**Liver cancer organoid culture**. Tumor organoid lines were generated from liver biopsy or resection tissue according to published protocols[8,9]. Briefly, tumor tissues were dissociated to small-cell clusters and seeded in domes of basement membrane extract type 2 (BME2, R&Dsystems, Cat. No. 3533-005-02). Polymerized BME2 domes were overlaid with expansion medium (EM): advanced DMEM/F-12 (Gibco, Cat. No. 12634010) supplemented with 1× B-27 (Gibco, Cat. No. 17504001), 1× N-2 (Gibco, Cat. No. 17502001), 10 mM Nicotinamide (Sigma, Cat. No. N0636), 1.25 mM N-Acetyl-L-cysteine (Sigma, Cat. No. A9165), 10 nM [Leu15]-Gastrin (Sigma, Cat. No. G9145), 10 µM Forskolin (Tocris, Cat. No. 1099), 5 µM A83-01 (Tocris, Cat. No. 2939), 50 ng/ml EGF (Peprotech, Cat. No. AF-100-15), 100 ng/ml FGF10 (Peprotech, Cat. No. 100-26), 25 ng/ml HGF (Peprotech, Cat. No. 100-39), 10% RSpo1-conditioned medium (v/v, homemade). Cultures were kept at 37 °C in a humidified 5% CO$_2$ incubator. Organoids were passaged weekly at 1:4–1:6 split ratios using 0.25% Trypsin-EDTA (Gibco, Cat. No. 25200056). Frozen stocks were prepared at regular intervals. All organoid cultures were regularly tested for Mycoplasma contamination using the MycoAlert™ Mycoplasma detection kit (Lonza, Cat. No. LT07-118).

**Histology and immunohistochemistry**. Tumor and liver tissues were fixed in 4% phosphate-buffered formalin and embedded in paraffin using standard procedures. Tumor organoids were released from BME2 by incubation in Dispase II (Sigma-Aldrich, Cat. No. D4693). Organoids were fixed in 4% phosphate-buffered formalin in PBS for 30 min at room temperature following encapsulation in HistoGel (Thermo Fisher Scientific, Cat. No. HG-4000-012) and subsequent dehydration and paraffin embedding.

Histopathological evaluation was assessed by three board-certified pathologists (MSM, JV and LMT). Tumors were classified based on architecture and cytological features, and graded according to the Edmondson grading system[10,11]. The following primary antibodies were used for automated diagnostic immunohistochemical staining on a Benchmark XT device (Ventana Medical Systems) at the Institute of Pathology of the University of Basel: AFP (Ventana, Ref-Nr. 760-2603), ARG1 (Ventana, Ref-Nr. 760-4801), CD10 (Ventana, Ref-Nr. 790-4506), CD56 (Ventana, Ref-Nr. 790-4465), CHGA (Ventana, Ref-Nr. 760-2519), GPC3 (Ventana, Ref-Nr. 790-4564), HLA-ABC (Abcam, Cat. No. ab70328), Hep Par-1 (Ventana, Ref-Nr. 760-4350), KRT19 (Ventana, Ref-Nr. 760-4281), Ki-67 (Dako, Cat. No. IR626), Pan-TRK (Abcam, Cat. No. ab181560), PD-L1 (Ventana, Ref-Nr. 740-4907), SYP (Ventana, Ref-Nr. 790-4407), and SSTR2 (Abcam, Cat. No. ab134152).

**Xenograft mouse model**. Experiments involving animals were performed in strict accordance with Swiss law and were previously approved by the Animal Care Committee of the Canton Basel-Stadt, Switzerland. Tumor organoids, corresponding to 2 × 10$^6$ cells, were released from BME2, resuspended in 100 µl 50:50 (v/v) BME2:expansion medium, and injected subcutaneously into the flank of one male NSG (Non-obese diabetic, Severe combined immunodeficiency, Gamma) mouse (The Jackson Laboratory) at 8 weeks of age. The mouse was housed in an individually ventilated cage (Tecniplast Green Line) at 22 °C, 55% humidity and a light cycle of 12:12 h. Tumor growth was assessed weekly by caliper measurement. The tumor was harvested when it reached 1000 mm$^3$ in size, fixed in 4% phosphate-buffered formalin and processed for paraffin embedding and immunohistochemistry as described above.

**Drug screenings**. All compounds were dissolved in DMSO at 10 mM (except for cisplatin and carboplatin) and aliquots were stored at −20 °C, 4 °C or room temperature according to the manufacturer's recommendations. Sorafenib tosylate, lenvatinib mesylate, cabozantinib mesylate, regorafenib, octreotide acetate, lanreotide acetate, etoposide, sunitinib malate, everolimus, entrectinib, larotrectinib: all from Selleckchem; pasireotide ditrifluoroacetate (MedChem Express); cisplatin (Sandoz); carboplatin (Labatec). For drug screenings, organoids were dissociated with 0.25% Trypsin-EDTA (Gibco) and seeded at 1000 cells/well in 384-well plates in organoid expansion medium supplemented with 10% BME2. Two days later, compounds were added in a 2-fold dilution series ranging from 0.02 nM to 10 µM. After 6 days of treatment, cell viability was measured using CellTiter-Glo 3D (Promega). Luminescence was measured on a Synergy H1 Multi-Mode Reader (BioTek Instruments). Results were normalized to vehicle control (100% DMSO or 100% water). All experiments were performed twice. Dose-response curves were calculated using Prism 9.3.1 (GraphPad), nonlinear regression algorithm was used with a constrain of 0 for the bottom and 100 for the top.

**DNA extraction and whole-exome sequencing**. DNA from the tumor, adjacent non-tumoral liver tissue and organoid was extracted using the Qiagen DNeasy Blood & Tissue kit (Qiagen, Cat. No. 69504) following the manufacturer's instructions. Extracted DNA was subjected to whole-exome sequencing. The Twist Human Core Exome kit was used for whole exome capture according to the manufacturer's guidelines. Sequencing was performed on Illumina NovaSeq 6000 using paired-end 100-bp (mean sequencing depth 135× for HCC, 153× for the organoid

and 129× for germline (adjacent non-tumoral liver tissue)). Sequencing was performed by CeGaT (Tübingen, Germany).

Reads obtained were aligned to the reference human genome GRCh38 using Burrows-Wheeler Aligner (BWA, v0.7.12)[12]. Local realignment, duplicate removal, and base quality adjustment were performed using the Genome Analysis Toolkit (GATK, v4.1 and Picard (http://broadinstitute.github.io/picard/)). Somatic single nucleotide variants (SNVs) and small insertions and deletions (indels) were detected using MuTect2 (GATK 4.1.4.1)[13] and Strelka (v.2.9.10)[14]. Only variants detected by both callers were kept. We filtered out SNVs and indels outside of the target regions (i.e., exons), those with a variant allelic fraction (VAF) of <5 % and/or those supported by <3 reads. We excluded variants for which the tumor VAF was <5 times that of the paired non-tumor VAF. We further excluded variants identified in at least two of a panel of 123 non-tumor samples, captured and sequenced using the same protocols using the artifact detection mode of MuTect2 implemented in GATK. All indels were manually inspected using the Integrative Genomics Viewer[15]. FACETS (v.0.5.14)[16] was used to identify allele-specific copy number alterations (CNAs). Genes with total copy number greater than gene-level median ploidy were considered gains; greater than ploidy + 4, amplifications; less than ploidy, losses; and total copy number of 0, homozygous deletions. Somatic mutations associated with the loss of the wild-type allele (i.e., loss of heterozygosity [LOH]) were identified as those where the lesser (minor) copy number state at the locus was 0. For chromosome X, the log ratio relative to ploidy was used to call deletions, loss, gains and amplifications. All mutations on chromosome X in male patients were considered to be associated with LOH. Comparison of copy number between organoids and tumor were performed at gene level.

**Reporting summary**. Further information on research design is available in the Nature Research Reporting Summary linked to this article.

## Results

**Case presentation**. A 74-year-old man presented with sudden one-sided loss of vision in the absence of other neurological symptoms. Diagnostic workup revealed giant cell arteritis (GCA) with involvement of the temporal artery, which was confirmed histologically and treated with aspirin as well as high-dose steroids. An imaging workup was initiated to investigate whether GCA was a paraneoplastic phenomenon. A computed tomography scan revealed a mass with maximum diameter of 4 cm in the gastric corpus as well as an intrahepatic mass with a maximum diameter of 4 cm. The intrahepatic mass was hypoechogenic in ultrasound and was highly suspicious for malignancy in the consecutive magnet resonance imaging (MRI) (Fig. 1a, b). Biopsies of the two lesions revealed two independent malignancies: A gastrointestinal stromal tumor (GIST) of the stomach and a poorly differentiated hepatocellular carcinoma (HCC). Liver values were within the normal range (aspartate aminotransferase (ASAT) of 17 U/l, alanine aminotransferase (ALAT) of 17 U/l, gamma-glutamyl transferase (GGT) of 30 U/l, alkaline phosphatase (AP) of 77 U/l). Liver function tests were normal, and the radiological findings were not suspicious for advanced liver fibrosis or cirrhosis. Tumor markers revealed high Alpha-Fetoprotein (AFP) levels of 754 kIU/l (normal range: <5.8 kIU/l), as well as slightly elevated carcinoembryonic antigen (CEA) (4.7 µg/l, normal range: <3.4 µg/l) and carboanhydrate-antigen 19-9 (CA 19-9) (47.7 U/ml, normal range: <34 U/ml) (Fig. 1a).

The interdisciplinary tumor board recommended a curative treatment approach with surgical resection. The patient underwent left-lateral liver resection and simultaneous partial gastrectomy without complications, allowing complete local resection of both tumors. Histopathological assessment confirmed the diagnosis of GIST of the stomach (Supplementary Fig. 1a). Based on the tumor size of 3.7 cm, mitotic activity and tumor localization, the GIST had a low risk for progression and required no further therapy[17]. Macroscopic evaluation of the resected liver tumor showed a polynodular tumor with beige and partly yellow cut-surface. Microscopy displayed a poorly differentiated tumor composed of medium to large cells with moderate to marked pleomorphism, growing in solid patternless sheets, lack of sinusoidal spaces and gland formation, compatible with Edmondson–Steiner grade IV HCC[10] (Supplementary Fig. 1b). Furthermore, lymphovascular and perineural invasion could be observed and the tumor was necrotic in about 20% (Supplementary Fig. 2). Immunophenotypic characterization resulted in the definitive diagnosis of poorly differentiated hepatocellular carcinoma (Edmondson-Steiner grade IV) with neuroendocrine differentiation (HCC-NED). The tumor cells were positive for Hep Par-1, Arginase 1 (ARG1), CD10, Glypican-3 (GPC3), and KRT19 (Fig. 1c, Supplementary Figs. 1 and 2). The same tumor cells were also positive for the neuroendocrine markers Synaptophysin (SYP) and Chromogranin (CHGA) (Fig. 1c and Supplementary Fig. 2). Weak positive staining for somatostatin-receptor 2 (SSTR2) was detected in 10% of the tumor cells (Supplementary Fig. 1c). In contrast CD56 was negative (Supplementary Fig. 1c). The proliferation marker KI-67 was expressed in 85% of the tumor cells, documenting a very high proliferation rate (Fig. 1c and Supplementary Fig. 2). The adjacent non-tumoral liver displayed no substantial alterations (Supplementary Figs. 1b and 2).

According to the recommendations of the interdisciplinary tumor board, the patient was enrolled into a postoperative HCC surveillance program. The first computed tomography (CT) scan three months after surgery revealed multifocal intrahepatic disease recurrence (max. diameter of 8 cm) with portal vein invasion as well as pulmonary metastasis, requiring palliative systemic therapy (Fig. 1a–c). Because HCC-NEDs are exceedingly rare, there was no published evidence to guide drug selection. The tumor board tentatively recommended drugs approved for the treatment of HCC but was well aware of the risk that the neuroendocrine differentiation of the tumor might limit the response to these drugs.

**Patient-derived HCC-NED organoids as a preclinical tumor model**. Within the last few years, patient-derived organoids have emerged as powerful preclinical model system to assess drug responsiveness of tumor cells in vitro, thereby allowing personalized medicine[18]. Accordingly, we generated organoids from the patient's resected HCC-NED tissue. HCC-NED organoids grew rapidly after initial seeding, allowing their expansion and characterization within a short time frame of 3 weeks (compared to an average model generation time of 8 weeks for HCC organoids[8]). Morphologically, HCC-NED organoids presented as solid spheroids comparably to other HCC organoid models[8]. Histologically, HCC-NED organoids retained growth pattern and differentiation grade of the original tumor (Fig. 1c). Notably, markers of hepatic (Hep Par-1) and neuroendocrine (SYP, CHGA) differentiation were equally retained in the organoids, further underlining their ability to recapitulate the tumor biology in vitro.

Moreover, to assess their in vivo tumorigenicity, HCC-NED organoids were subcutaneously injected in immunodeficient mice. HCC-NED organoids could indeed easily be propagated as xenografts. The xenografts retained histological features as well

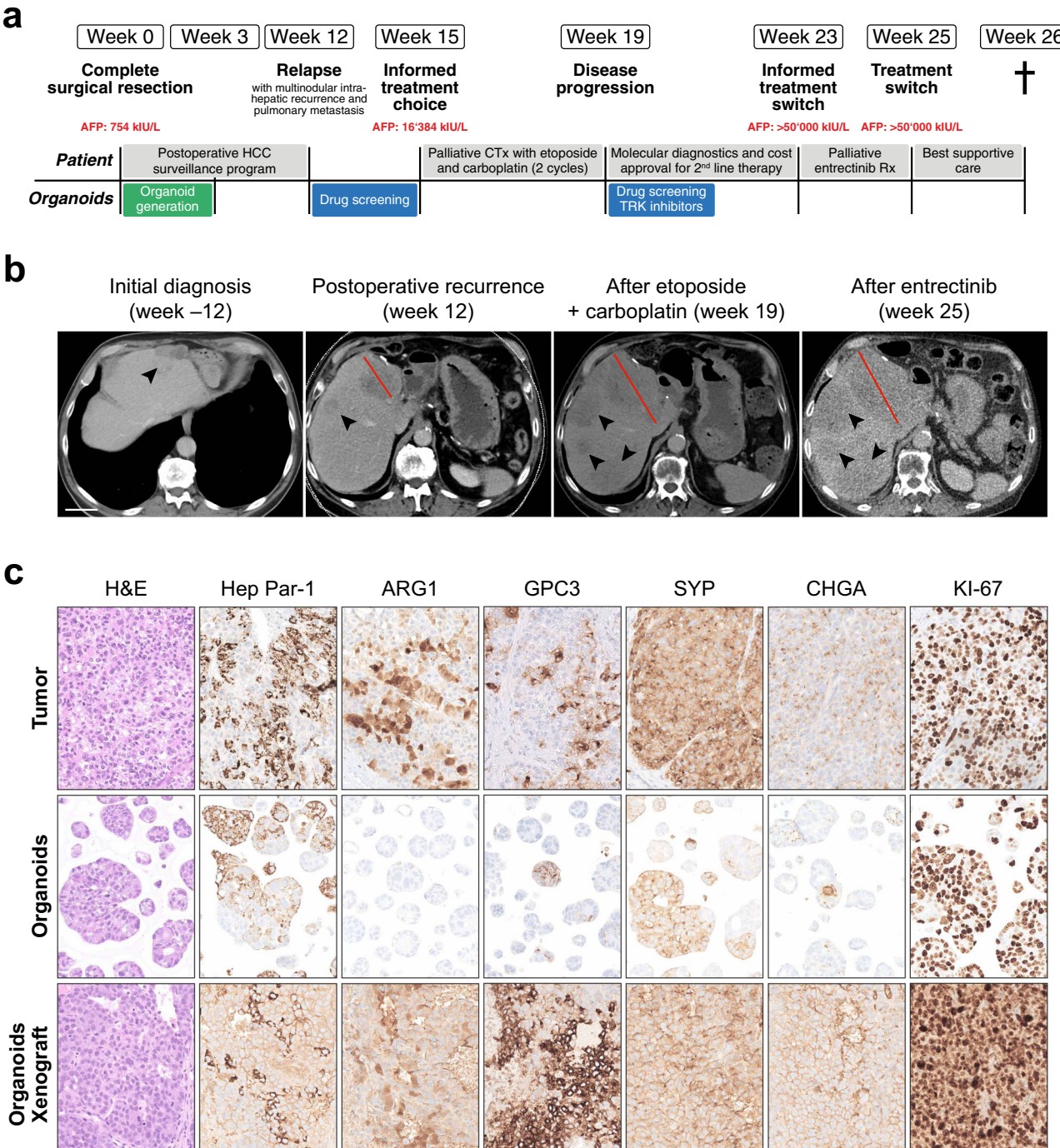

**Fig. 1 Poorly differentiated hepatocellular carcinoma with neuroendocrine differentiation. a** Clinical time course of the patient. AFP Alpha Fetoprotein, CTx chemotherapy, HCC hepatocellular carcinoma, Rx treatment. **b** Representative computed tomography (CT) scans of the abdomen with contrast at diagnosis (12 weeks before surgical resection), after postoperative recurrence (week 12), after two cycles of etoposide and carboplatin therapy (week 19) and after two weeks of entrectinib treatment (week 25). Transverse plane in the portal venous phase. The tumor lesion marked in red increased in size from 6.9 cm (week 12) to 10.8 cm after systemic chemotherapy (week 19) and measured 9.4 cm after entrectinib treatment (week 25). Scale bar: 5 cm. **c** Histopathological characteristics of the resected primary tumor as well as the matched organoids and organoid-derived xenograft (representative images of the microscopic findings). Immunohistochemical stainings were performed for hepatocellular (Hep Par-1, ARG1) as well as neuroendocrine markers (SYP, CHGA). Scale bar: 100 μm. ARG1 Arginase 1, CHGA Chromogranin A, GPC3 Glypican-3, H&E hematoxylin and eosin, Hep Par-1 Hepatocyte Paraffin 1, SYP Synaptophysin.

as marker expression reminiscent of the patient's primary tumor, including the very high expression rate of the proliferation marker KI-67 (Fig. 1c and Supplementary Fig. 1c).

**HCC-NED displays mutations in *TP53*, *CTNNB1* and *NTRK1*.** It has been shown that the genetic profile can influence clinical decision-making and treatment selection in several cancer types[19]. Therefore, we performed whole-exome sequencing (WES) of the HCC-NED tumor (mean coverage 136×) matched to the non-tumoral liver tissue (mean coverage 130×) and its derived organoids (mean coverage 154×) to identify targetable alterations.

We detected 111 and 116 somatic mutations in the HCC-NED tumor and HCC-NED organoids, respectively (Supplementary Data 1). Of those, 106 were shared between both samples (87.6%; Fig. 2a). Moreover, analysis of genome-wide copy number alterations detected by WES showed a 71 % correlation between HCC-NED tumor and HCC-NED organoid including the loss of chr 5, 8p, 1q, gain of chr 13p, 20p and the focal amplification of the 19q12 locus (Supplementary Fig. 3a).

The genomic analysis revealed that the HCC-NED tumor harbored *CTNNB1* (p.S45P) and *TP53* (p.R273C) hotspot mutations (Fig. 2b). Both genes are frequently mutated in HCC[20]. TP53 is also commonly mutated in gastroenteropancreatic neuroendocrine cancers[21]. Furthermore, *NTRK1*, encoding the Neurotrophic Receptor Tyrosine Kinase 1, was found to harbor a missense variant (p.T741P) of unknown significance (VUS) in the tyrosine kinase (TK) domain in tumor and matched organoids (Fig. 2b, Supplementary Fig. 3b and Supplementary Data 1).

**HCC-NED organoids are sensitive towards carboplatin, etoposide and entrectinib.** Because HCC with neuroendocrine differentiation is a very rare entity, treatment allocation is still unclear and relies on documentations found in single case reports. As a first treatment option for this patient, the combination of atezolizumab and bevacizumab was considered, because it is the current first-line therapy for advanced HCC[20,22]. However, immunostaining revealed the lack of Human Leucocyte Antigen (HLA) ABC and Programmed Death-Ligand 1 (PD-L1) expression on tumor cells (Supplementary Fig. 1c, e), and therefore, the efficacy of immune checkpoint inhibitor therapy might be impaired in this patient[23–25]. Current guidelines recommend a limited number of systemic treatments for advanced HCC or for NET/NEC[21,26] (Supplementary Fig. 4a). To identify potentially effective drugs, we used the HCC-NED organoids for in vitro drug response testing. 4 HCC organoids from our biobank (Supplementary Data 2) served as controls. Drug screenings were performed with a broad drug dilution range of 0.02 nM to 10 µM. Cells were treated for six days at which time the cell number was then determined using an ATP-based readout as described in the materials and methods. HCC-NED organoids showed the same response to the multikinase-inhibitors sorafenib, lenvatinib, cabozantinib and regorafenib as the control HCC organoid lines (Fig. 2c, Supplementary Fig. 4b, Supplementary Data 3 and 4). We then tested drugs approved for advanced neuroendocrine tumors and carcinomas, including somatostatin analogs (octreotide, lanreotide, pasireotide), the multikinase-inhibitor sunitinib, the mTOR-inhibitor everolimus as well as conventional chemotherapeutics (5-FU, cisplatin, etoposide, carboplatin). Compared to conventional HCC organoids, HCC-NED organoids responded better to the classical chemotherapeutics cisplatin, etoposide and carboplatin (Fig. 2c, d, Supplementary Fig. 4c, Supplementary Data 3 and 4). Of note, no antitumoral activity could be observed for any of the somatostatin analogs. We also tested the pan-TRK inhibitors entrectinib and

larotrectinib. In a recent report, these drugs were strong growth inhibitors of organoids derived from gastroenteropancreatic neuroendocrine neoplasms[27]. Larotrectinib had no effect on our HCC-NED and HCC organoids (Fig. 2c, Supplementary Fig. 4d and Supplementary Data 4). On the other hand, HCC-NED organoids were growth inhibited by entrectinib with an IC50 of 0.61 µM (Fig. 2c, d, Supplementary Data 3 and 4).

**Progressive disease after two cycles of etoposide and carboplatin, followed by a treatment attempt with entrectinib.** Based on the above-mentioned results, a first-line palliative chemotherapy with etoposide and carboplatin was initiated. The therapy was well tolerated by the patient. However, a CT scan after 4 weeks revealed progressive disease (Fig. 1b), and the treatment was stopped after only two cycles. Because next-generation sequencing analysis of the tumor had revealed a *NTRK1*-mutation and the pan-TRK inhibitor entrectinib was effective in HCC-NED organoids, a therapy attempt with entrectinib was suggested for second line. After patient consultation, interdisciplinary discussion at the tumor board, and approval by the health insurance company, entrectinib treatment was initiated 4 weeks later. Unfortunately, the patient's general health condition had rapidly deteriorated and he additionally suffered a traumatic femoral neck fracture. Entrectinib treatment was still initiated but had to be discontinued after two weeks. A CT scan revealed progression of the number and size of the metastatic lesions, but also increasing areas without up-take of contrast material (non-viable tumor) (Fig. 1b). A formal evaluation of the response to entrectinib was not possible due to the short treatment period. The patient then received best supportive care. He deceased one week later.

**Discussion**

Hepatocellular carcinoma with neuroendocrine differentiation is a very rare tumor entity[1,2]. No evidence-based treatment options are established for these aggressive primary liver carcinomas. Published reports describe the use of surgical resection, percutaneous ablation, transarterial chemoembolization and classical systemic chemotherapies[4]. However, most reports describe a poor outcome despite these treatments. In the present report, we describe for the first time a functional precision oncology approach to guide the choice of systemic chemotherapeutics in liver cancer. We successfully generated tumor organoids that were then used for testing the efficacy of drugs in vitro. The HCC-NED organoid line grew very rapidly, probably reflecting the exceedingly high proliferation rate of the originating tumor. It is known that high proliferation rates increase the success rate for generating organoids from tumor biopsies[8]. We therefore believe that HCC-NECs and HCC-NEDs are good candidates for generating tumor organoids. Furthermore, the rapid growth of HCC-NED organoids allowed their characterization as well as drug testing within a time frame of 5–6 weeks, the latter being an important factor when using pre-clinical models for therapy guidance. Indeed, the applicability of tumor organoid models in the clinical setting strongly depends on the time scale of establishment[28–30].

In our case, despite good efficacy in the organoid tumor model in vitro, the combination of etoposide and carboplatin as first-line palliative therapy was clinically not effective. We can only speculate about the reasons for this failure. Clearly, the organoid models, while maintaining most of the key cellular and molecular features of the originating tumors, have important limitations because they lack the tumor stroma. This precludes testing anti-angiogenic drugs such as ramucirumab or immune-checkpoint inhibitors. It is also conceivable that the tumor stroma influences the response to treatments targeted to the tumor cells. Such

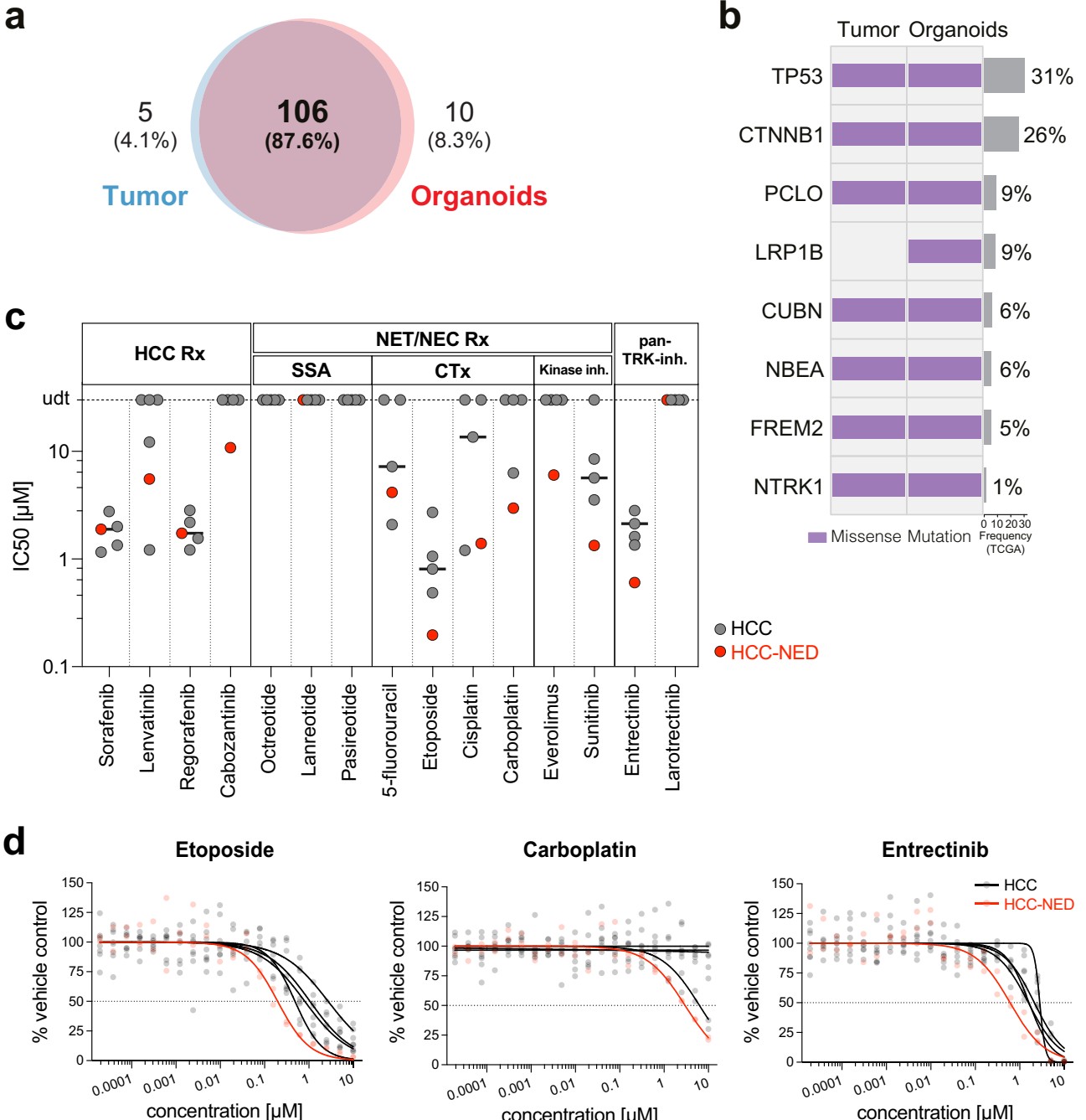

**Fig. 2 Genetic characterization of HCC-NED and drug screening using patient-derived organoids. a** Venn diagram representing the number of somatic mutations detected in each sample using whole-exome sequencing (WES). **b** Oncoprint of genetic alterations detected in the HCC-NED tumor and its matched organoids by WES. Alterations are colored according to the legend. Alterations shown are those included in the cancer gene list (Supplementary Data 1). **c** Half maximal inhibitory concentration (IC50) as determined in the HCC-NED (colored in red) and four different HCC organoid lines (colored in black). Horizontal lines indicate the median IC50. CTx chemotherapy, HCC hepatocellular carcinoma, HCC-NED hepatocellular carcinoma with neuroendocrine differentiation, inh. inhibitors, NEC neuroendocrine carcinoma, NET neuroendocrine tumor, Rx therapy, SSA somatostatin analogs. **d** Dose-response curves for the chemotherapeutics etoposide and carboplatin as well as the pan-TRK inhibitor entrectinib with concentrations ranging from 0.02 nM to 10 μM. The HCC-NED organoid line is colored in red, the four HCC organoid lines included as reference in black. After 6 days of treatment, an ATP-based readout was used as a surrogate for cell number. All values were normalized to vehicle control (DMSO or water) and are displayed as the mean of n = 2 biologically independent experiments. DMSO dimethyl sulfoxide.

effects cannot be assessed in the organoid models. It is also possible that the drug concentrations in the tumor were too low, or that the tumors developed rapid resistance to etoposide and carboplatin in vivo.

Fusions involving *NTRK* genes are the most common mechanisms of oncogenic TRK activation[31]. Typically, the fusions contain 3' sequences of *NTRK*s that include the kinase domain and 5' sequences of a different gene. The fusion results in a

chimeric oncoprotein with ligand-independent constitutive activation of the TRK kinase[32]. Clinical detection of *NTRK* fusions is mainly based on next-generation sequencing (NGS). Immunohistochemistry is a complementary method that can detect TRK overexpression as a surrogate for *NTRK* fusions[33,34]. The missense mutation present in our case (T741P) was not associated with TRK overexpression, since both HCC-NED tissue and organoids stained negative in pan-TRK IHC (Supplementary Fig. 1d).

An increasing number of *NTRK* mutations and splice variants and cases of TRK ectopic expression and/or overexpression has been reported[32]. However, overall *NTRK1* is not frequently mutated in neuroendocrine tumors such as pancreas[35,36] and prostate[37] (0 and 1.2% frequency, respectively), in HCC the frequency is 0.7%[38–43]. For most of these *NTRK* mutations, the functional consequences and their role as oncogenic drivers are unknown or remain controversial. In our case, we have no evidence that the mutation in the kinase domain is indeed a driver mutation. The observation that HCC-NED organoids did not respond to the selective pan-TRK inhibitor larotrectinib, but were sensitive to entrectinib, a pan-TRK inhibitor with additional activity against the proto-oncogene kinase ROS1 and anaplastic lymphoma kinase (ALK)[44] does not support the hypothesis that constitutive TRK activity was a main oncogenic driver in our case. Moreover, *NTRK1* (T741P) was predicted to be deleterious by the MetaSV score[45]. Of note, this is only based on in silico predictions and further studies are required to unveil the functional impact of this mutation.

Limited resources did not allow us to test all conventional chemotherapeutics listed in the guidelines for NET/NEC treatment[21]. Specifically, we did not investigate the antitumoral efficacy of temzolomide, streptozocin, capecitabine, leucovorin, oxaliplatin and irinotecan in our HCC-NED organoid model. Accordingly, we cannot rule out that these cytostatic drugs might have displayed antitumoral activity. The efficacy of these components can be tested to potentially inform future treatment decisions. Furthermore, this unique HCC-NED organoid line can be used to screen additional drug libraries, an effort that might identify promising candidates for future cases of HCC-NED.

In conclusion, we describe a rare case of a patient with HCC with neuroendocrine differentiation. The rapid establishment of patient-derived tumor organoids allowed in vitro drug testing and thereby personalized treatment choices in this patient. Unfortunately, the drugs could not prevent the rapid tumor progression. Nevertheless, the report provides a first proof-of-principle for using organoids for personalized medicine in these rare primary liver cancers.

## Data availability

The WES data reported here are available under restricted access at the European Genome- Phenome Archive under primary accession number EGAS00001005887. Access is restricted because genetic data is personally identifiable. To obtain access and conditions of access to the EGA datasets, contact the corresponding authors, who will respond within 4 weeks. The use of the data will be subjected to agreement of a data use policy, which details the minimum protection measures required related to data encryption and user access. The data will be available to the authorized users for the duration of the requested project. Users will have to specifically agree to preserve, at all times, the confidentiality of information and Data pertaining to Data Subjects and to use or attempt to use the Data to compromise or otherwise infringe the confidentiality of information on Data Subjects and their right to privacy. User have to agree not to attempt to identify Data Subjects. The full data use policy will be available upon data access request. Source data for Fig. 2 panels a and b can be found in Supplementary Data 1. Source data for Fig. 2 panel c can be found in Supplementary Data 3. Source data for Fig. 2 panel d, and Supplementary Fig. 4 panels b, c, and d, can be found in Supplementary Data 4.

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

## Acknowledgements

We thank the patient for study participation, the operating room caretakers for the support in collecting biological material and Prof. Dr. Katharina Rentsch for serum AFP measurement. We thank Dr. Xueya Wang for the support with animal experiments and Petra Hirschmann for performing immunohistochemical stainings. Furthermore, we are very grateful for the support of Dr. Diego Calabrese and his team at the Histology Core Facility of the Department of Biomedicine at the University of Basel. Entrectinib and larotrectinib were a kind gift of Prof. Dr. Alfred Zippelius. This work was funded by European Research Council Synergy grant 609883 (MERiC) and by SystemsX.ch grant MERiC to M.H.H. S.P. was supported by the Swiss Cancer League (KFS-4988-02-2020-R), by the Theron Foundation, Vaduz (LI), by the Surgery Department of the University Hospital Basel and by the Prof. Max Clöetta Stiftung. M.A.M. was supported by a personal grant (MD-PhD-4500-06-2018) of the Swiss Cancer Research Foundation. L.M.T. was supported by AIRC grant number IG 2019 Id.23615. The funding bodies had no role in study design; in the collection, analysis and interpretation of data; in the writing of the report; and in the decision to submit the article for publication.

## Author contributions

M.A.M., S.N., M.C.L., S.P. and M.H.H. conceived the study and designed the experiments; R.D., S.D.S., O.K., D.B. and M.H.H. recruited the patient and were involved in patient care; M.S.M., C.E., J.V. and L.M.T. performed histopathological analyses of the resected tissue; M.A.M., S.N., M.S.M., M.C.L. and J.G. performed experiments and analyzed the data; M.A.M., S.N., M.C.L., M.S.M., S.P. and M.H.H. wrote the manuscript; M.H.H. and S.P. obtained funding.

## Competing interests

The authors declare no competing interest.
