## [Peer Review File · Communications Medicine]

Reviewers' comments:

Reviewer #1 (Remarks to the Author):

In this manuscript from the Heim lab, the authors present a case report of a liver cancer with neuroendocrine differentiation (HCC-NEDs). The authors characterise the tumour and also generate tumour organoids that they show recapitulate aspects of the tumour of the patient in a dish, in particular its differentiation status. Then, they use these for a drug screening to identify a potential treatment for this patient.

Overall, the novelty of this paper resides in the generation of this very rare tumour subtype and the practise of using the organoids to inform patient treatment. Although unfortunately the patient condition was too poor to pursue that further, this is a very interesting point that demonstrates the applicability of the system and the feasibility of the approach, highly debated in terms of timeframe. For that, this manuscript deserves attention. Of note, I think that concept should be strengthened in the manuscript as the authors only touch on it at the end of the discussion and I have the impression that it would be also important for general audience.

Having said that, the manuscript suffers from many errors on figure callings, and missing data and requires revision in that respect. The missing Xcell spreadsheets with the mutations have generated extra unnecessary questions would it have been provided. It has impeded the evaluation of that part of the manuscript. I would urge the authors to pay attention to these details.

Comments:

1) There is some sort of confusion with the figure callings between Figure 1, S1 and S2:

1.1) Line 140 Microscopy displayed a poorly differentiated tumor compatible with Edmondson-Steiner grade IV HCC with solid growth pattern and lack of gland formation (Fig. S1B). This is not clear from the image presented. The authors should show the scoring that lead to that diagnostic or at least the features that were used to grade the tumour like that.

1.2) line 142: lymphovascular and perineural invasion could be observed and the tumor was necrotic in about 20%. The authors should add the call for figure S2

1.3) Immunophenotypic characterization resulted in the definitive diagnosis of poorly differentiated hepatocellular carcinoma (Edmondson-Steiner grade IV) with neuroendocrine differentiation (HCC-NED). The tumor cells were positive for Hep Par-1, Arginase 1 (ARG1), CD10, and Glypican-3 (GPC3), but negative for the biliary marker KRT7 (Fig 1B, Fig. S2).

First, authors should also call here Fig S1C

Second, in the text reads "KRT7" but the figure reads KRT19, both in Figure S1C and figure S2.

-Line 149 Weak positive staining for somatostatin-receptor 2 (SSTR2) was detected in 10% of the tumor cells. Here authors should also call Fig S1C.

-Line 153 displayed no significant alterations (Fig. S1B) here should be Fig S2.

-In line 212 it should read Table S2

- In the table of contents of supplementary materials the Supplementary Figure 4 is missing

2) Genetic mutations recapitulated in the organoids.

Basically, that part is impossible to assess. The authors mention that supplementary Table S1, containing the list of somatic mutations will be provided as spreadsheet, however this is nowhere to be found in the materials, so this section of the manuscript cannot be evaluated. Similarly, in the text the authors mention (line 189) that the WES showed a 71% correlation between tumour and organoid (fig S3A). However, the figure does not show a correlation number, just the correlation between the different types of mutations between tissue and culture. How do the authors obtain this 71%?

Along these same lines, the authors attribute specific mutations to CTNNB1 and TP53 pS45P and R273C, but this information, again, is nowhere to be found in the figure.

Finally, the authors comment on NTRK1 mutation. Is this present in the organoid and tissue or only in the tissue? This information is also missing from text, figure and/or figure legend.

Overall, these points should be fixed before considering the manuscript for publication.

Minor point:

I would swap figure 1C for Figure 1A, to increase flow and readability

Reviewer #2 (Remarks to the Author):

This case report details a patient presenting with sudden vision loss as part of a paraneoplastic syndrome associated with gastric and intrahepatic masses. The liver mass was found to be a poorly differentiated carcinoma with expression of hepatocellular (Heppar1, arginase, CD10) and neuroendocrine markers (synaptophysin, chromogranin, CD10, CK19-) without evidence of biliary differentiation by cytokeratin 7 stain. Notably the proliferation index 85% (ki-67) and serum AFP, 754, was markedly elevated. The background liver was non-cirrhotic. The patient underwent curative resection which was followed by multifocal recurrence of the malignancy at 3 months. Tumor organoids were generated and tested for drug sensitivity. The organoids retained expression of Heppar1 and synapto. and chromo. IHC markers. These organoids easily generated subcutaneous xenografts. The patient received treatment based upon the organoids in vitro sensitivity to etoposide and carboplatin. Genomic analysis revealed an NTRK1 mutation and the patient received entrectinib for 2 weeks. The tumor progressed but there was evidence of necrosis on imaging.

Comments:

This is an exceedingly rare tumor with practically no guidance for therapy. Everything presented, IHC pattern, proliferation index, serum AFP, lack of cirrhosis and histology support this is indeed a hepatocellular carcinoma with neuroendocrine differentiation. There is so little in the literature about this tumor it is a bit surprising the authors missed the other reports on this tumor such as Aboelenen 2014, possibly Barsky 1984, Garcia 2006, and Yang 2009 among others.

There needs to be discussion of NTRK1 mutations in other neuroendocrine tumors, such as large cell NET of the lung.

Why not reach out to find more of this tumor type and look for NTRK1 in additional cases?

Particularly since this missense mutation is not a driver or, according to the authors data, associated with TRK over expression.

Were the xenografted mice treated with entrectinib? Did these show any evidence of response? Interestingly the authors state “immunostaining revealed the lack 207 of human leucocyte antigen (HLA) ABC expression on tumor cells (Fig. S1C), and 208 therefore, immune checkpoint inhibitor therapy was not further considered.” Not sure if this approach should be lauded or discouraged. This is a very forward and aggressive approach. This makes theoretic sense however this probably needs to be stricken from the report. This is not a diagnostic test. Why wasn't PD-L1 expression investigated? While PD-L1 expression is not an accepted test for HCC, it reasonably correlates with response to checkpoint inhibitor (CPI) therapy. This information might dissuade others with HCC-NED from exploring CPI.

Tremendous to learn these tumors might harbor NTRK-1 mutations. Disappointing the evidence is unclear on response. On the one hand there is a suggestion there is tumor necrosis on imaging but there was unequivocal tumor progression. Perhaps the treatment was not long enough and the patient's general state of health was poor. Even more disappointing no post mortem, even abdomen / liver alone, analysis was possible.

Reviewer #3 (Remarks to the Author):

This is a very interesting and well documented case of a HCC-NED, which is extremely rare and there is no established systemic standard treatment. The prognosis is usually fatal. This case will help the community to further guide systemic treatment decisions. Concerning the NTRK kinase mutation, I am in doubt, whether this is really a driver mutation. To further help to understand this, Markus Heim and colleagues should quote the allelic frequency of the found mutation and also give database evidence that it is pathogenic. If the allelic frequency is low, I would be more critical - especially in the abstract - concerning the meaning of this mutation. I recommend also to elaborate why lack of HLA ABC expression was used as argument not to give Atezo/Bev. Based on figure 2C, it seems to me that the multikinase inhibitors SOR and REGO seems to be equally effective in comparison to Eto/Platin. Please elaborate a little bit on this.

Manuscript: COMMSMED-21-0467

Patient-derived tumor organoids for personalized medicine in a rare case of hepatocellular carcinoma with neuroendocrine differentiation

Marie-Anne Meier, Sandro Nuciforo, Mairene Coto-Llerena, John Gallon, Matthias S. Matter, Caner Ercan, Jürg Vosbeck, Luigi M. Terracciano, Savas D. Soysal, Daniel Boll, Otto Kollmar, Raphaël Delaloye, Salvatore Piscuoglio, and Markus H. Heim

REVIEWER COMMENTS

Introductory remarks

We thank the reviewers for their constructive suggestions that helped us to improve this manuscript. In this new version of our manuscript, we have addressed all major points raised by the reviewers. We would like to point out that figure and table callings were adapted to match the journal's style, e.g. "Figure Sx" and "Table Sx" were renamed to "Supplementary Figure x" and "Supplementary Table x", respectively. All changes in text and figures have been integrated in the manuscript as outlined in our detailed point-by-point reply below.

Reviewer #1 (Remarks to the Author):

In this manuscript from the Heim lab, the authors present a case report of a liver cancer with neuroendocrine differentiation (HCC-NEDs). The authors characterise the tumour and also generate tumour organoids that they show recapitulate aspects of the tumour of the patient in a dish, in particular its differentiation status. Then, they use these for a drug screening to identify a potential treatment for this patient.

Overall, the novelty of this paper resides in the generation of this very rare tumour subtype and the practise of using the organoids to inform patient treatment. Although unfortunately the patient condition was too poor to pursue that further, this is a very interesting point that demonstrates the applicability of the system and the feasibility of the approach, highly debated in terms of timeframe. For that, this manuscript deserves attention. Of note, I think that concept should be strengthened in the manuscript as the authors only touch on it at the end of the discussion and I have the impression that it would be also important for general audience.

Having said that, the manuscript suffers from many errors on figure callings, and missing data and requires revision in that respect. The missing Xcell spreadsheets with the mutations have generated extra unnecessary questions would it have been provided. It has impeded the evaluation of that part of the manuscript. I would urge the authors to pay attention to these details.

Reply: we thank the Reviewer for the appreciation of our work. We deeply apologize for not having uploaded Supplementary Table 1 to the editorial manager during our initial submission. We now provide the missing table with all relevant and detailed information regarding our WES analysis. We appreciate the Reviewer's comment about the feasibility of our approach in the reported timeframe. Indeed, this represents a current challenge in the organoid field, since most tumor organoid models rely on establishment times of several weeks to months. We now strengthen this point in the manuscript as follows:

Results (line 133)

"HCC-NED organoids grew rapidly after initial seeding, allowing their expansion and characterization within a short time frame of 3 weeks (compared to an average model generation time of 8 weeks for HCC organoids)"

Discussion (line 235)

"Furthermore, the rapid growth of HCC-NED organoids allowed their characterization as well as drug testing within a time frame of 5-6 weeks, the latter being an important factor when using pre-clinical models for therapy guidance. Indeed, the applicability of tumor organoid models in the clinical setting strongly depends on the time scale of establishment"

Comments:

1) There is some sort of confusion with the figure callings between Figure 1, S1 and S2:

Reply: we apologize for the confusion. We now corrected all missing callings as outlined below.

1.1) Line 140 Microscopy displayed a poorly differentiated tumor compatible with Edmondson-Steiner grade IV HCC with solid growth pattern and lack of gland formation (Fig. S1B). This is not clear from the image presented. The authors should show the scoring that lead to that diagnostic or at least the features that were used to grade the tumour like that.

Reply: we apologize for the lack of information. We now describe in more detail the features that were used to grade the tumor according to Edmondson and Steiner¹.

Results (line 101)

“Microscopy displayed a poorly differentiated tumor composed of medium to large cells with moderate to marked pleomorphism, growing in solid patternless sheets, lack of sinusoidal spaces and gland formation, compatible with Edmondson-Steiner grade IV HCC”

1.2) line 142: lymphovascular and perineural invasion could be observed and the tumor was necrotic in about 20%. The authors should add the call for figure S2

Reply: done.

1.3) Immunophenotypic characterization resulted in the definitive diagnosis of poorly differentiated hepatocellular carcinoma (Edmondson-Steiner grade IV) with neuroendocrine differentiation (HCC-NED). The tumor cells were positive for Hep Par-1, Arginase 1 (ARG1), CD10, and Glypican-3 (GPC3), but negative for the biliary marker KRT7 (Fig 1B, Fig. S2). First, authors should also call here Fig S1C Second, in the text reads “KRT7” but the figure reads KRT19, both in Figure S1C and figure S2.

Reply: we now also call Supplementary Figure 1c. “KRT7” was corrected to “KRT19” in the results section.

- Line 149 Weak positive staining for somatostatin-receptor 2 (SSTR2) was detected in 10% of the tumor cells. Here authors should also call Fig S1C.

Reply: done.

- Line 153 displayed no significant alterations (Fig. S1B) here should be Fig S2.

Reply: the non-tumoral liver tissue displayed in Supplementary Figure 1b is also adjacent to the tumor tissue, in addition to that we now also refer to Supplementary Figure 2, as correctly noticed by the reviewer.

- In line 212 it should read Table S2

Reply: done.

- In the table of contents of supplementary materials the Supplementary Figure 4 is missing

Reply: we now added the missing information and updated the page numbers in the table of contents.

2) Genetic mutations recapitulated in the organoids. Basically, that part is impossible to assess. The authors mention that supplementary Table S1, containing the list of somatic mutations will be provided as spreadsheet, however this is nowhere to be found in the materials, so this section of the manuscript cannot be evaluated.

Reply: we apologize for not having uploaded Supplementary Table 1. The missing table can now be found as attachment to our resubmission.

Similarly, in the text the authors mention (line 189) that the WES showed a 71% correlation between tumour and organoid (fig S3A). However, the figure does not show a correlation number, just the correlation between the different types of mutations between tissue and culture. How do the authors obtain this 71%?

Reply: we apologize for the lack of clarity. The correlation among the copy number was done at gene level. This is now better described in the methods section as follows:

Supplementary Methods (line 105)

“Comparison of copy number between organoids and tumor were performed at gene level”

Along these same lines, the authors attribute specific mutations to CTNNB1 and TP53 pS45P and R273C, but this information, again, is nowhere to be found in the figure.

Reply: we apologize for the missing information. Supplementary Table 1 now lists all specific mutations found in tumor tissue and matched organoids.

Finally, the authors comment on NTRK1 mutation. Is this present in the organoid and tissue or only in the tissue? This information is also missing from text, figure and/or figure legend.

Reply: Figure 2b shows the presence of the NTRK1 mutation in both tumor and matched organoids. To better clarify this point, we now changed the text accordingly and also refer to the relevant figures and tables as follows:

Results (line 164)

Furthermore, NTRK1, encoding the Neurotrophic Receptor Tyrosine Kinase 1, was found to harbor a missense variant (p.T741P) of unknown significance (VUS) in the tyrosine kinase (TK) domain in tumor and matched organoids (Fig. 2b, Supplementary Figure 3b and Supplementary Table 1).

Overall, these points should be fixed before considering the manuscript for publication.

Minor point:

I would swap figure 1C for Figure 1A, to increase flow and readability

Reply: we now modified the figure as requested.

Reviewer #2 (Remarks to the Author):

This case report details a patient presenting with sudden vision loss as part of a paraneoplastic syndrome associated with gastric and intrahepatic masses. The liver mass was found to be a poorly differentiated carcinoma with expression of hepatocellular (Heppar1, arginase, CD10) and neuroendocrine markers (synaptophysin, chromogranin, CD10, CK19-) without evidence of biliary differentiation by cytokeratin 7 stain. Notably the proliferation index 85% (ki-67) and serum AFP, 754, was markedly elevated. The background liver was non-cirrhotic. The patient underwent curative resection which was followed by multifocal recurrence of the malignancy at 3 months. Tumor organoids were generated and tested for drug sensitivity. The organoids retained expression of Heppar1 and synapto. and chromo. IHC markers. These organoids easily generated subcutaneous xenografts. The patient received treatment based upon the organoids in vitro sensitivity to etoposide and carboplatin. Genomic analysis revealed an NTRK1 mutation and the patient received entrectinib for 2 weeks. The tumor progressed but there was evidence of necrosis on imaging.

Comments:

This is an exceedingly rare tumor with practically no guidance for therapy. Everything presented, IHC pattern, proliferation index, serum AFP, lack of cirrhosis and histology support this is indeed a hepatocellular carcinoma with neuroendocrine differentiation.

There is so little in the literature about this tumor it is a bit surprising the authors missed the other reports on this tumor such as Aboelenen 2014, possibly Barsky 1984, Garcia 2006, and Yang 2009 among others.

Reply: indeed, we did not cite Aboelen 2014, Barsky 1984 as well as Yang 2009. Aboelen 2014 and Yang 2009 both describe liver tumors consisting of >99% NEC (in H&E being monotonous, small- to medium- sized cells) and only very few (<1%) tumor cells expressing hepatocellular markers (e. g. Hep Par-1). Taken together, these two cases describe primarily NECs in the H&E staining with a small cell fraction being positive for hepatocellular markers in IHC. We accordingly decided to omit those citations, as in our case the diagnosis of HCC was made based on histomorphology, rather than solely IHC marker expression. The case report described by Barsky and colleagues 1984 was not cited, as the carcinoid tumor could not be further characterized using immunostaining for neuroendocrine markers, but rather based on ultrastructural features in electron microscopy. The reference Garcia 2006, however, is included in our Supplementary Fig. 4, where we describe previous treatment options used in patients with similar neuroendocrine liver malignancies.

There needs to be discussion of NTRK1 mutations in other neuroendocrine tumors, such as large cell NET of the lung. Why not reach out to find more of this tumor type and look for NTRK1 in additional cases? Particularly since this missense mutation is not a driver or, according to the authors data, associated with TRK over expression.

Reply: we thank the reviewer for the suggestion. In the revised version of the manuscript, we explored the frequency of NTRK1 mutation in other neuroendocrine tumors. *NTRK1* was not mutated in a cohort of 108 neuroendocrine pancreatic tumors^{2,3} and mutated with a frequency of 1.2% in a cohort of 114 neuroendocrine prostate tumors⁴. Additionally, we checked the frequency of *NTRK1* mutation in other HCC samples⁵⁻¹⁰. In HCC, *NTRK1* was found to be mutated in 7 of 1007 samples.

Figure for the Reviewers:

Legend: Lollipop plot showing the mutational frequency of NTRK1 in neuroendocrine tumors and HCC (figure was generated from cBioportal).

In the revised version of the manuscript, we have added this information as follows:

Discussion (line 265)

“NTRK1 is not frequently mutated in other neuroendocrine tumors such as pancreas and prostate (0 and 1.2% frequency, respectively) and in HCC the frequency is 0.7%”

Were the xenografted mice treated with entrectinib? Did these show any evidence of response?

Reply: after establishing the first generation of xenografts, we did not further expand the xenograft cohort because of the rapid deterioration of the patient’s general condition.

Interestingly the authors state “immunostaining revealed the lack 207 of human leucocyte antigen (HLA) ABC expression on tumor cells (Fig. S1C), and 208 therefore, immune checkpoint inhibitor therapy was not further considered.” Not sure if this approach should be lauded or discouraged. This is a very forward and aggressive approach. This makes theoretic sense however this probably needs to be stricken from the report. This is not a diagnostic test.

Reply: low or absent HLA-ABC expression has been associated with impaired response to immune checkpoint inhibitors (ICI) in solid tumors^{11,12}. However, indeed results are conflicting and other parameters (e.g. PD-L1 expression, WNT-signaling activity as well as a combination of serum AFP and CRP levels) have been suggested as predictive parameters for response to ICI therapy¹³⁻¹⁶. We have modified the text accordingly:

Results (line 174)

“However, immunostaining revealed the lack of human leucocyte antigen (HLA) ABC and programmed death-ligand 1 (PD-L1) expression on tumor cells, and therefore, the efficacy of immune checkpoint inhibitor therapy might be impaired in this patient”

Why wasn't PD-L1 expression investigated? While PD-L1 expression is not an accepted test for HCC, it reasonably correlates with response to checkpoint inhibitor (CPI) therapy. This information might dissuade others with HCC-NED from exploring CPI.

Indeed, we agree this is an important point to address. Accordingly, we now additionally show that PD-L1 staining was negative (Supplementary Fig. 1E).

Tremendous to learn these tumors might harbor NTRK-1 mutations. Disappointing the evidence is unclear on response. On the one hand there is a suggestion there is tumor necrosis on imaging but there was unequivocal tumor progression. Perhaps the treatment was not long enough and the patient's general state of health was poor. Even more disappointing no post mortem, even abdomen / liver alone, analysis was possible.

Reply: indeed, deterioration of the patients' general condition following entrectinib treatment precluded additional assessments of response to treatment. Unfortunately, autopsy was declined by the patient's family. Therefore, we were not able to analyze the tumor post-mortem.

Reviewer #3 (Remarks to the Author):

This is a very interesting and well documented case of a HCC-NED, which is extremely rare and there is no established systemic standard treatment. The prognosis is usually fatal. This case will help the community to further guide systemic treatment decisions.

Reply: we thank the reviewer for the appreciation of our work.

Concerning the NTRK kinase mutation, I am in doubt, whether this is really a driver mutation. To further help to understand this, Markus Heim and colleagues should quote the allelic frequency of the found mutation and also give database evidence that it is pathogenic. If the allelic frequency is low, I would be more critical - especially in the abstract - concerning the meaning of this mutation.

Reply: we thank the reviewer for the insightful comments. We have now included this information in Supplementary Table 1. *NTRK1* mutation was found to have an allelic frequency of 0.55 (in the tumor) and a cancer cell fraction (CCF) of 0.93 suggesting this mutation is clonal. Regarding its pathogenicity, *NTRK1* T741P mutation was predicted deleterious by METASV score¹⁷. However, these are only *in silico* predictions and further studies are required to unveil the functional impact of this mutation. In the revised version of the manuscript, we now add this as limitation as follows:

Discussion (line 274)

“Moreover, NTRK1 (T741P) was predicted to be deleterious by the MetaSV score¹³. Of note, this is only based on in silico predictions and further studies are required to unveil the functional impact of this mutation”

We also changed the abstract to clarify the unknown significance of the NTRK1 mutation.

Abstract (line 43)

“Because genomic analysis revealed a NTRK1-mutation of unknown significance (kinase domain) and tumor organoids were sensitive to entrectinib, a pan-TRK inhibitor, the patient received entrectinib as second line therapy.”

I recommend also to elaborate why lack of HLA ABC expression was used as argument not to give Atezo/Bev.

Reply: previous reports suggested that low or absent HLA ABC expression might impair response to immune checkpoint inhibitors, however indeed results are conflicting^{11,12}. Other parameters such as

PD-L1 expression or CRP levels together with serum AFP have however recently been shown to allow prediction of response to immunotherapy in HCC¹³⁻¹⁵. We have now added Supplementary Fig. 1E to show that PD-L1 staining was negative, and we have changed the text in the Result section and added the corresponding references.

Results (line 174)

“However, immunostaining revealed the lack of human leucocyte antigen (HLA) ABC and programmed death-ligand 1 (PD-L1) expression on tumor cells, and therefore, the efficacy of immune checkpoint inhibitor therapy might be impaired in this patient.”

Based on figure 2C, it seems to me that the multikinase inhibitors SOR and REGO seems to be equally effective in comparison to Eto/Platin. Please elaborate a little bit on this.

Reply: the fact that HCC organoids displayed more variability in their response to classic chemotherapies (5-FU, etoposide and both platinum compounds) as compared to targeted therapies such as sorafenib and regorafenib can be due to multiple reasons such as:

- Differences in their mode of action: Classical chemotherapeutics (as compared to kinase inhibitors) have a different target entity (DNA vs. protein), therefore tumor-intrinsic activity of DNA repair pathways might affect response to classical chemotherapeutics, however not kinase inhibitors¹⁸.
- Differences in cellular uptake mode: Hepatocellular uptake of sorafenib (and probably also regorafenib) has been suggested to occur mainly via passive diffusion¹⁹⁻²¹. Uptake of classical chemotherapeutics however has been suggested to be more dependent on the expression of cell membrane transporters^{22,23}. Taken together, cellular uptake of classical chemotherapeutics (as compared to the uptake of sorafenib) might therefore be more dependent on the intrinsic expression levels of membrane transporters.

REFERENCES

1. Edmondson, H.A. & Steiner, P.E. Primary carcinoma of the liver: a study of 100 cases among 48,900 necropsies. *Cancer* **7**, 462-503 (1954).
2. Scarpa, A., *et al.* Whole-genome landscape of pancreatic neuroendocrine tumours. *Nature* **543**, 65-71 (2017).
3. Jiao, Y., *et al.* DAXX/ATRX, MEN1, and mTOR pathway genes are frequently altered in pancreatic neuroendocrine tumors. *Science* **331**, 1199-1203 (2011).
4. Beltran, H., *et al.* Divergent clonal evolution of castration-resistant neuroendocrine prostate cancer. *Nat Med* **22**, 298-305 (2016).
5. Pilati, C., *et al.* Genomic profiling of hepatocellular adenomas reveals recurrent FRK-activating mutations and the mechanisms of malignant transformation. *Cancer Cell* **25**, 428-441 (2014).
6. Harding, J.J., *et al.* Prospective Genotyping of Hepatocellular Carcinoma: Clinical Implications of Next-Generation Sequencing for Matching Patients to Targeted and Immune Therapies. *Clinical cancer research : an official journal of the American Association for Cancer Research* **25**, 2116-2126 (2019).
7. Schulze, K., *et al.* Exome sequencing of hepatocellular carcinomas identifies new mutational signatures and potential therapeutic targets. *Nat Genet* **47**, 505-511 (2015).
8. Ahn, S.M., *et al.* Genomic portrait of resectable hepatocellular carcinomas: implications of RB1 and FGF19 aberrations for patient stratification. *Hepatology* **60**, 1972-1982 (2014).
9. Fujimoto, A., *et al.* Whole-genome sequencing of liver cancers identifies etiological influences on mutation patterns and recurrent mutations in chromatin regulators. *Nat Genet* **44**, 760-764 (2012).
10. Cancer Genome Atlas Research Network. Electronic address, w.b.e. & Cancer Genome Atlas Research, N. Comprehensive and Integrative Genomic Characterization of Hepatocellular Carcinoma. *Cell* **169**, 1327-1341 e1323 (2017).
11. Rodig, S.J., *et al.* MHC proteins confer differential sensitivity to CTLA-4 and PD-1 blockade in untreated metastatic melanoma. *Sci Transl Med* **10**(2018).
12. Lee, J.H., *et al.* Transcriptional downregulation of MHC class I and melanoma de-differentiation in resistance to PD-1 inhibition. *Nat Commun* **11**, 1897 (2020).

13. Scheiner, B., *et al.* Prognosis of patients with hepatocellular carcinoma treated with immunotherapy - development and validation of the CRAFTY score. *J Hepatol* (2021).
14. Sangro, B., *et al.* Association of inflammatory biomarkers with clinical outcomes in nivolumab-treated patients with advanced hepatocellular carcinoma. *J Hepatol* **73**, 1460-1469 (2020).
15. Zhu, A.X., *et al.* Pembrolizumab in patients with advanced hepatocellular carcinoma previously treated with sorafenib (KEYNOTE-224): a non-randomised, open-label phase 2 trial. *The Lancet Oncology* **19**, 940-952 (2018).
16. Sia, D., *et al.* Identification of an Immune-specific Class of Hepatocellular Carcinoma, Based on Molecular Features. *Gastroenterology* **153**, 812-826 (2017).
17. Kim, S., Jhong, J.H., Lee, J. & Koo, J.Y. Meta-analytic support vector machine for integrating multiple omics data. *BioData Min* **10**, 2 (2017).
18. Martin, L.P., Hamilton, T.C. & Schilder, R.J. Platinum resistance: the role of DNA repair pathways. *Clinical cancer research : an official journal of the American Association for Cancer Research* **14**, 1291-1295 (2008).
19. Swift, B., *et al.* Sorafenib hepatobiliary disposition: mechanisms of hepatic uptake and disposition of generated metabolites. *Drug Metab Dispos* **41**, 1179-1186 (2013).
20. Hu, S., *et al.* Interaction of the multikinase inhibitors sorafenib and sunitinib with solute carriers and ATP-binding cassette transporters. *Clinical cancer research : an official journal of the American Association for Cancer Research* **15**, 6062-6069 (2009).
21. Haralampiev, I., *et al.* The interaction of sorafenib and regorafenib with membranes is modulated by their lipid composition. *Biochim Biophys Acta* **1858**, 2871-2881 (2016).
22. More, S.S., *et al.* Organic cation transporters modulate the uptake and cytotoxicity of picoplatin, a third-generation platinum analogue. *Mol Cancer Ther* **9**, 1058-1069 (2010).
23. Girardi, E., *et al.* A widespread role for SLC transmembrane transporters in resistance to cytotoxic drugs. *Nat Chem Biol* **16**, 469-478 (2020).

REVIEWERS' COMMENTS:

Reviewer #1 (Remarks to the Author):

The authors have answered all my concerns. I want to congratulate the authors on the re-writting of the organoid section. It is very pleasing to read the section of the timeline from organoid establishment to drug testing. I think this strengthens the manuscript and also helps the community. Finally, I have reviewed the missing Table 1 with the mutations and associated text and I have no concerns on that either. I believe the manuscript is ready for publication.

I have no further comments

Reviewer #2 (Remarks to the Author):

The authors have addressed my concerns adequately. Thank you.

Reviewer #3 (Remarks to the Author):

Well done. Everything is adressed